



# Verification of the mixed layer depth in the OceanMAPS operational forecast model

Daniel Boettger[1], Robin Robertson[2] and Gary B. Brassington[3]

[1]School of Mathematics and Statistics, University of New South Wales, Sydney, 2052, Australia
[2]Xiamen University Malaysia, Selangor Darul Ehsan, 43900 Sepang, Malaysia
[3]Bureau of Meteorology, Sydney, 2000, Australia

*Correspondence to*: Daniel Boettger (d.boettger@student.unsw.edu.au)

**Abstract.** The ocean mixed layer depth is an important parameter describing the exchange of fluxes between the atmosphere and ocean. In ocean modelling a key factor in the accurate representation of the mixed layer is the parameterisation of
vertical mixing. An ideal opportunity to investigate the impact of different mixing schemes was provided when the Australian Bureau of Meteorology upgraded its operational ocean forecasting model, OceanMAPS to version 3.0. In terms of the mixed layer, the main difference between the old and new model versions was a change of vertical mixing scheme from that of Chen et al to the General Ocean Turbulence Model.

The model estimates of the mixed layer depth were compared with those derived from Argo observations. Both versions of
the model exhibited a deep bias in tropical latitudes and a shallow bias in the Southern Ocean, consistent with previous studies. The bias however, was greatly reduced in version 3.0, and variance between model runs decreased. Additionally, model skill against climatology also improved significantly. Further analysis discounted changes to model resolution outside of the Australian region having a significant impact on these results, leaving the change in vertical mixing scheme as the main factor in the assessed improvements to mixed layer depth representation.

**1 Introduction**

The mixed layer depth (MLD) is an important factor controlling the dynamics of air-sea interaction. As a proxy for the heat content of the ocean boundary layer, the MLD is critical in understanding the exchange of heat and moisture fluxes between the ocean and atmosphere. It also has biological consequences, with the depth of the mixed layer having a critical impact on primary productivity.

Because of its importance to air-sea interaction, an accurate representation of the MLD has been long considered a key dynamic of climate models. It is equally important in ocean general circulation models, particularly those used for operational ocean forecasting. The MLD plays a role in determining the likelihood of atmospheric convection over the ocean, has been linked to the development of severe weather in mid-latitude cyclones (Chambers et al., 2015), and is also a key factor in the development and intensity of tropical cyclones  (Mao et al., 2000;Zhao and Chan, 2017).

The Ocean Modelling and Prediction System (OceanMAPS), the operational ocean forecasting system of the Australian Bureau of Meteorology (BoM), transitioned from version 2.2.1 to version 3.0 on 11 April 2016. While a number of changes were introduced in version 3.0, the most significant in terms of the MLD was a change in vertical mixing scheme from a modified version of Chen et al. (1994) scheme to the General Ocean Turbulence Model (GOTM, Burchard et al., 1999). During the transition period, both versions continued to run in parallel until July 2016, utilising the same observational data
and atmospheric forcing. This provided an ideal opportunity to verify two versions of an operational forecasting system under essentially identical conditions. In this case, by isolating other factors, a verification of each model version also provided insights into the performance of each vertical mixing parameterisation in an operational setting.

The purpose of this study is therefore (1) to quantify the impact of changes to the OceanMAPS forecasting system on the estimation of the MLD, and (2) to assess the performance of the different vertical mixing parameterisations. These results





will then inform future development of OceanMAPS. Pertinent details of OceanMAPS, plus a description of the observational data used, are in Section 2. Section 3 details the method used for the calculation of the MLD, while the results of the analysis are reported in Section 4. Finally, the key results are discussed and expanded upon in Section 5.

## 2 Data

### 2.1 The model

The BoM has run the OceanMAPS operational ocean forecasting model since 2007. The main components of this system are the Ocean Forecasting Australian Model (OFAM), a data assimilation system, and atmospheric forcing from BoM's ACCESS-G model (Puri et al., 2013). While the atmospheric forcing remains unchanged between version 2.2.1 and version 3.0, changes to both OFAM and BODAS that will impact the calculation of the MLD are highlighted below, with full details

in Table 1.

The OFAM is a near-global (polar regions are excluded), eddy-resolving implementation of the Modular Ocean Model version 4p1 (MOM 4p1, Griffies, 2009). The latest version, OFAM3, is described in Oke et al. (2013). While the Chen et al. (1994) vertical mixing scheme had been used in OFAM2 (OceanMAPS version 2.2.1), the OFAM3 model implemented in OceanMAPS version 3.0 uses the General Ocean Turbulence Model (GOTM, Burchard et al., 1999). The Chen et al. (1994)

scheme is a hybrid between a traditional bulk layer and the dynamical instability model of Price et al. (1986). It has been widely used in climate studies, particularly in tropical regions. Being initially formulated as an explicit MLD model, it was modified for use in the MOM by Power et al. (1995). The GOTM, conversely, is an attempt to unify many of the well-known turbulence closure schemes into a single model, with the characteristics of individual models replicated by changing the values of a number of constants.

In version 3.0, GOTM is configured as a $k$-$\varepsilon$ scheme. Following Rodi (1987), the rate of turbulent dissipation is calculated by:

$$\frac{D\varepsilon}{Dt} = \mathcal{D} + \frac{\varepsilon}{k}(c_{\varepsilon 1}S + c_{\varepsilon 3}G - c_{\varepsilon 2}\varepsilon) \tag{1}$$

where $\mathcal{D}$ is the sum of the viscous and turbulent transport terms, $S$ and $G$ are the rates of shear and buoyancy production, and $c_{\varepsilon *}$ are model constants. The constant $c_{\varepsilon 3}$ was defined such that if the buoyancy was positive (upwards), $c_{\varepsilon 3}$ was equal to zero. This resulted in zero buoyancy production in cases where the buoyancy profile was convectively unstable. As $G$ is generally

an order of magnitude smaller than $S$, it only plays a significant role in turbulent mixing when $G$ is relatively large and $S$ relatively small. The impacts of these settings on the results are discussed further in section 5.1.

Another major change in OFAM3 was an increase in horizontal resolution. OFAM2 employed a telescopic horizontal grid, with a 0.1° resolution around Australia (16°N to 75°S, 90°E to 180°E) gradually decreasing outside of this region. In OFAM3, horizontal resolution was fixed at 0.1° throughout the entire domain. While this can be expected to result in a

marked improvement in the estimation of the MLD outside of the Australian region; within this region the impact on MLD will only been seen towards the boundaries, where the accuracy of incoming fluxes is improved. To isolate the effect of changing vertical mixing parameterisations, the analysis in this study is limited to the region where both models provided 0.1° resolution.

Version 2.2.1 ran four independent model cycles on consecutive days, with the spin-up period starting nine days before the

forecast start. Primarily designed to minimise over-fitting of the model fields to the available observations, this arrangement also enables the generation of a lagged time ensemble. In version 3.0, the number of model cycles was reduced to three, with the spin-up period extending only six days. With initial verification of version 3.0 indicating a resultant significant decrease in sea surface temperature error (Bureau of Meteorology, 2017), it is probable that this will also have a positive impact on MLD estimation.

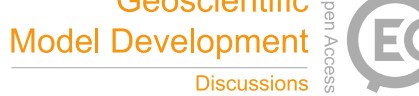

## 2.2 The dataset

Both the model and observational data were sourced from the Class 4 dataset (Ryan et al., 2015), developed through the GODAE OceanView program in order to allow direct comparison of member organisation's ocean forecast model against a single Argo temperature and salinity dataset. For OceanMAPS, the model 24 h mean temperature and salinity fields are

interpolated (nearest-neighbour in the horizontal, linearly in the vertical) onto the Argo profiles out to +144 h. The dataset also includes a corresponding climatological temperature and salinity profile (Boyer et al., 2013), as well as a persistence forecast, where the model forecast is compared with a succession of the best estimate profiles from the previous six model runs.

Class 4 data for the period 11 April 16 to 4 July 16 were used, covering the overlap of OceanMAPS versions 2.2.1 and 3.0.

Each profile was inspected to remove any unrealistic values or vertical gradients (Johnson et al., 2013), and profiles were only used in the analysis if a MLD was identified in the observed profile and in each model profile. The resultant quality controlled dataset provided 5316 individual profiles over the area of interest.

## 3. Calculation of the Mixed Layer Depth

Conceptually, the MLD is well understood to represent the depth over which the mixing of surface fluxes has occurred. But

the large variety of definitions used in the literature demonstrate the difficulty in accurately determining the MLD in all situations. Noting the relative paucity of in-situ salinity observations, the majority of these identify the depth at which the temperature varies from the near-surface temperature by a certain amount, usually between 0.2 and 1.0 °C. As seawater density in the mid-latitudes and tropics is mostly proportional to temperature, this method generally provides a good estimate of the depth of the pycnocline and hence the depth to which surface mixing is limited. The minor dependence of density on

salinity does, however, become important in some cases, particularly in the presence of a *barrier layer* or *compensating layer*.

Large amounts of precipitation, a common occurrence in equatorial regions, can result in a layer of cool, less saline water at the ocean surface overlaying a warmer, saltier layer. In this case, the surface isopycnal layer will be thicker than the isothermal layer, with the difference between the two termed the *barrier layer* (Lukas and Lindstrom, 1991). While a

different mechanism is responsible, this type of vertical profile is also observed in high latitudes over winter months, where cool ocean surface temperatures overlay relatively warm, subsurface water (Kara et al., 2000). In the presence of a barrier layer, the temperature profile is a poor proxy for the depth of the mixed layer, and the density profile should be used.

Another surface mixed layer scenario is typified by a temperature and salinity profile that each have a negative gradient over the same depth, at such a rate that the density remains constant. Termed a *compensating layer*, this commonly occurs in

regions with mean annual negative Ekman pumping (such as at the centre of subtropical gyres) and within subtropical convergence zones during winter (de Boyer Montégut et al., 2004). Although the density gradient is small in this instance, mixing is inhibited by the large temperature and salinity gradients present; consequently, the MLD is best defined as the top of the thermocline.

With salinity profiles available for both the model and observational datasets used here, the calculation of density is

straightforward, and both a temperature and a density threshold can be used to account for the scenarios discussed above. Following de Boyer Montégut et al. (2004), the MLD is therefore defined as the first depth at which either of the following criteria are met:

$$\Theta = \Theta_{ref} \pm 0.2 \tag{2}$$

$$\rho_\Theta = \rho_{\Theta ref} + 0.03 \tag{3}$$

Potential temperature, $\Theta$, and potential density, $\rho_\Theta$, are used to negate the depth dependence of the thermal expansion coefficient. The subscript *ref* denotes the value of each parameter at the reference depth, here set at a depth of 10 m in order





to avoid diurnal variation that may be present in the observations but not reproduced in the daily mean model profiles. Temperature inversions are accounted for by using an absolute difference in Eq. (2). Using the shallowest depth derived by either the temperature or the density criterion ensures that the correct criteria is selected in both barrier layer and compensating layer scenarios.

In 58 % of Argo profiles both the temperature and density criteria are met within 10 m or 5 % of the MLD (Fig. 1) and the locations where a single criterion has been used shows general agreement with previous studies. A compensating layer (i.e. where the temperature criterion is satisfied first) identified near Tasmania has been previously reported to exist during the winter months (de Boyer Montégut et al., 2004;Schiller and Ridgway, 2013), and locations of barrier layers (i.e. the density criterion is satisfied first) near the Equator and in the Southern Ocean correspond with regions of high precipitation and

relatively cool sea surface temperatures respectively. These results afford confidence that in most occasions the MLD is being correctly identified with this method.

## 4 Results

### 4.1 Observed and forecast mixed layer depth

The MLD determined from the Argo observations (Fig. 2) matches the general trends for this season seen in previous studies

(Carton et al., 2008;Kara et al., 2003;de Boyer Montégut et al., 2004;Schiller and Ridgway, 2013;Holte et al., 2016), and are in line with the conceptual model of seasonal mixed layer dynamics. In the tropics, the MLD is almost uniformly of the order of 20 m, with the small range of values indicated by the 5[th] and 95[th] percentiles shown at Fig. 2a. During the austral autumn, the inter-tropical convergence zone shifts northwards from northern Australia towards the Equator, resulting in weak momentum and heat fluxes into the ocean. Deeper mixed layers are seen in the sub-tropical latitudes, particularly over the

Coral Sea; here the south easterly trade wind regime generates increasingly strong winds and subsequently increases ocean mixing (Fig. 2b). Higher variability is also expected here due to the mesoscale structure of the coastal boundary currents. The deepest mixed layers are seen south of Australia, with values approaching 250 m around 50°S, where the Antarctic Circumpolar Current (ACC) is on average most active (Rintoul and Sokolov, 2001).

While the model results exhibit a similar spatial trend, the zonal mean of each model (Fig. 3a) identifies some distinct biases.

Both models over-predict the depth of the mixed layer in the region 20°S - 40°S, and under-predict the MLD around 45°S - 65°S. A comparison between model versions shows that the magnitude of these biases has been reduced in version 3.0, while in the region 45°S - 50°S the bias has disappeared. This can be attributed to a number of distinctly deeper estimations of the MLD in the region of the ACC to the south of Australia (Fig. 3 b and c). In addition, the variability of the zonal mean between model forecast times (Fig. 3a) has decreased at all latitudes in version 3.0. This could be indicative of the changes

to the forecast cycle implemented in version 3.0. By reducing the spin up period and the number of individual ensemble members, the variability in the observational and atmospheric model forcing between each member is also reduced.

### 4.2 Model error

To provide a more quantitative assessment of the differences between each model version, the magnitude of the difference between the observed and forecast MLD was calculated (Fig. 4). The regional bias discussed previously is again evident

when the difference between each model and the Argo observations is plotted (Fig. 4 b and c), with both models forecasting a deeper MLD in mid-latitudes and a shallower MLD south of Australia. As could be expected, the largest errors are seen in regions of high mesoscale activity, such as the East Australian Current, the Leeuwin Current, and the ACC.

The mean absolute error (MAE, Fig. 4a) highlights the differences between model versions, particularly in the Southern Ocean. Here, the MAE has been reduced in version 3.0 by around 10 m, with a correspondingly large decrease in the spread

of the error, indicated by magnitude of the 95[th] percentile. North of 20°S the improvement in MAE is more modest, but

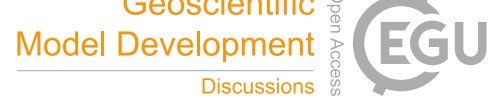



throughout the domain version 3.0 performs consistently better than version 2.2.1. The one exception to this trend; however, occurs around 30°S, where a spike in the MAE of version 3.0 occurs. More in-depth analysis reveals that this can be attributed to the area between 90°E and 100°E, where version 3.0 shows significantly larger errors than version 2.2.1 (Fig. 5, circled).

### 4.3 Model skill

A more quantitative measure of the forecasting ability of a model is the skill score (SS), here defined as the ratio of the root means square error (RMSE) of the model and a reference dataset (Ryan et al., 2015):

$$SS = 1 - \left( \frac{RMSE[model]}{RMSE[reference]} \right) \qquad (4)$$

A positive skill score indicates that the model is a better predictor of the future state of the ocean then the reference dataset, while a negative skill score indicates the opposite. Commonly, climatology is used as a reference, with typical skill scores for operational ocean forecast models in the range 0.2 to 0.7 for parameters such as temperature, salinity and sea surface height.

The Class 4 dataset includes monthly temperature and salinity fields (Boyer et al., 2013) interpolated to the location and date of each Argo observation. Typically, the climatological skill score of a model decreases with forecast lead time (i.e. the model is less skilful looking further into the future), and this trend is seen in both versions of OceanMAPS (Fig. 6, light blue and red). To increase the number of profiles contained in each bin, here the data has been separated into tropical (16°N - 20°S), mid-latitude (20°S - 45°S) and high latitude (45°S - 70°S) regions. Both models are more skilful than climatology within that data limits (+144 h), and in each region, there is a distinct improvement in version 3.0 compared to 2.2.1.

The skill score also objectifies the relative difficulty of forecasting the state of the ocean in different regions. For example, in tropical waters there is relatively little spatial and temporal variation in the ocean boundary layer, and so it is more difficult for the model to make a significant improvement over the climatology. In the mid-latitudes, where mesoscale features not resolved by the climatology are dominant, larger skill scores are seen.

Model skill can also be measured against a persistence forecast, where each model time step is compared against the best estimate field for that model run. Persistence skill scores typically increase with increasing lead time; as time increases the current estimate of the ocean becomes a less useful estimate of its future state. Persistence skill scores for each version of OceanMAPS are difficult to interpret, with no clear trend for either model version (Fig. 6, dark blue and red). In the tropics for instance, the negative skill scores suggest that it is more often useful to rely on the persistence field then the actual forecast field. A possible cause for the irregular persistence results is the OceanMAPS forecast cycle, in which three (version 3.0) or four (version 2.2.1) independent model runs are initiated on consecutive days. Under this arrangement, the persistence scores are comparing forecast fields from different runs that have been forced with different observational datasets, introducing another layer of variability.

### 5 Discussion

### 5.1 Southern Ocean response to model changes

The greatest improvements from version 2.2.1 to version 3.0, in terms of absolute error, occurred in the Southern Ocean and in particular around the ACC. One possible cause for this is the adoption of a global 0.1° horizontal grid in version 3.0 – in version 2.2.1 a telescopic grid was used, with 0.1° resolution around Australia and an expanding resolution elsewhere.

In the case of a telescopic grid, the benefits of increasing resolution are generally not seen close to the boundaries of the high-resolution region; it takes a certain number of grid cells to transform the low-resolution representation of the incoming fluxes into an accurate high-resolution field. Consequently, if the MAE is zonally averaged over the latitudes of the ACC,





there should be a decrease in MAE from 90°E and 180°E (the edges of the high-resolution region) towards the centre of the domain. Conversely, with a uniform horizontal resolution and in the absence of other factors, version 3.0 should exhibit a zonally uniform MAE.

Examining Fig. 5 however, the zonal MAE in the region of the ACC is generally uniform for both model versions, so a

change in global resolution does not appear to have had a significant effect on the estimation MLD. As a result, it is likely that changing the mixing scheme to GOTM is responsible for the improved results in this region.

South of the ACC, in the region 55°S to 65°S, the improvements seen in version 3.0 are less apparent. Here the zonal mean MAE is of a similar magnitude for each model version (Fig. 4), with version 2.2.1 outperforming version 3.0 in some areas (Fig. 5). If a change of mixing scheme is responsible for the significant improvements seen elsewhere, then the reason that

improvements are not seen here may lay in the distinct stratification profile common in high latitudes.

In version 3.0, buoyant production of turbulent kinetic energy, $G$, was limited to zero in instances where the buoyancy was positive (section 2.1). The impacts of this on the MLD can be determined by examining the net surface fluxes of both buoyancy $F_G$ and shear $F_S$. These are defined by:

$$F_G = \frac{-g\alpha Q_0}{\rho c_p} + g\beta(E-P)S_0 \tag{5}$$

$$F_S = \left(\frac{\tau}{\rho}\right)^{3/2} \frac{1}{\kappa z} \tag{6}$$

In Eq. (5), $\alpha$ and $\beta$ are the thermal expansion and haline contraction coefficients, $Q_0$ is the net surface heat flux, $E$ and $P$ the

total evaporation and precipitation, and $S_0$ the sea surface salinity. In Eq. (6), $\tau$ is the surface wind stress, $\kappa$ von Karman's constant and $z$ set to the upper-most layer of the model (2.5 m).

The zonal mean of the $F_G$ and $F_S$ from OceanMAPS version 3.0 are shown in Fig. 7. In the region of the ACC, $F_G$ is negative and stabilises the mixed layer, while the surface wind stress generates a large $F_S$ that drives turbulent mixing and generates a deep mixed layer. Conversely, south of 60°S, $F_G$ is positive and $F_S$ relatively small. Here, the $G$ term in Eq. (1) would be

significant, and the fact that this has been set to zero within version 3.0 will have a large impact on MLD estimates. This is the likely cause for the relatively poor performance of version 3.0 compared to version 2.2.1 seen in Fig. 5. A further conclusion that can be drawn from the Southern Ocean results is that version 3.0 significantly outperforms version 2.2.1 in shear-dominated mixing regions, whereas this improvement is negated in those regions where convective overturning is a significant factor.

**5.2 Impact of the MLD definition on results**

While the criteria used to identify the MLD are identical for both observation and model profiles, the individual characteristics of these datasets can result in varying levels of sensitivity to the same thresholds. For example, as the much greater vertical resolution of the Argo profiles allow finer features to be captured, it is possible that there will be some cases where a shallow temperature or density gradient not resolved in the model results in an apparent over-forecasting of the

MLD. While a MLD definition based on simple temperature and density thresholds is effective for a typical mixed layer profile consisting of a single isothermal layer above the thermocline, more complex profiles can be incorrectly characterised. To investigate what impact the magnitude of the temperature and salinity thresholds may have on results, a sensitivity study was conducted in which the thresholds were varied by up to ± 50 % (Fig. 8). The impact is intuitive; an increase (decrease) in the magnitude of the thresholds increases (decreases) the estimated MLD. However, the effect is non-linear, with the

observed MLD 12 % shallower when the threshold is 50 % smaller, but only 7 % deeper when the threshold is 50 % larger. Interestingly, OceanMAPS version 2.2.1 is more sensitive, and version 3.0 less sensitive, to threshold changes than the observations.





This disparity between model versions can be explained by examining individual profiles, with a typical example shown in Fig. 8 (right). Here, the version 3.0 profile appears to capture the shape of the observed profile better than version 2.2.1. Critically, the version 3.0 profile is very well-mixed above the MLD, whereas the observations and the version 2.2.1 profile exhibit slight temperature and density gradients in this region, which have triggered the MLD thresholds. It is these gradients that control the sensitivity of the dataset to changes in the threshold magnitude.

The stratification of the layer above the MLD can be quantified using the standard deviation of the potential temperature and potential density (Fig. 9); a low value indicates a well-mixed layer while a high value indicates stratification. The observations reveal that the greatest amount of stratification above the MLD exists in the tropical regions north of 15°S, and that density exhibits a higher degree of spatial variability than temperature. A comparison with the model results explains the biases present in OceanMAPS; both versions produced a more well-mixed layer than observed in the mid-latitudes (where Fig. 3 indicated a deep bias) and a more stratified layer in the Southern Ocean (where Fig. 3 indicated a shallow bias). The differences in mixing between model versions is also evident here; while the GOTM mixing scheme used in version 3.0 generally produces a uniformly isothermal surface layer, the Chen et al. (1994) scheme used in version 2.2.1 often produces a more stratified layer that is more susceptible to changes in threshold magnitudes.

The difference between the sensitivity of the observations and each model version then raises the question: how does the choice of threshold affect the measurement of error in each model? The answer to this exhibits a strong spatial dependence. Decreasing (increasing) the magnitude of the thresholds decreases (increases) the MAE north of 15°S, but increases (decreases) the MAE south of this point (Fig. 10a, for threshold changes of ± 50 %). Clearly, an optimal MLD definition may include spatially varying thresholds; however, unless carefully implemented this may introduce other errors. Instead, this information is best used as an indication of areas where a comparison between model versions can be given more or less credence. For example, version 2.2.1 is more sensitive to the choice of thresholds in the Southern Ocean, but has a similar sensitivity to version 3.0 elsewhere.

One region where this general trend is not followed is the box 20 – 30 °S, 90 – 100 °E, where version the 3.0 MAE is more sensitive to threshold changes than version 2.2.1. This is the same region that was highlighted in section 0, where version 3.0 had a larger MAE than version 2.2.1. Analysis of individual profiles in this region (e.g. Fig. 8) reveal a number of instances where the Argo and OceanMAPS version 2.2.1 profiles exhibit weak temperature and density gradients that trigger the MLD thresholds, whereas the OceanMAPS version 3.0 profile is well-mixed. In this case, the higher sensitivity shown in Fig. 10 for version 3.0 exists because, while changing the threshold has a small impact on the version 3.0 MLD, it has a large impact on the observed MLD that is subsequently expressed as 'error'. Finally, some inferences may be drawn on the vertical mixing schemes in each model version; in areas where weak stratification is present within the mixed layer, the GOTM scheme in version 3.0 produces a deeper mixed layer than the Chen et al. (1994) scheme in version 2.2.1. This also suggests that the simplification of the buoyancy term in the GOTM implementation is not producing sufficient negative buoyancy to stabilise and stratify the mixed layer. However, in most cases the improvements to the shear-generated mixing still result in version 3.0 outperforming version 2.2.1.

## 6 Conclusion

The ability of version 2.2.1 and version 3.0 of the OceanMAPS operational ocean forecast model to accurately resolve the mixed layer depth (MLD) was quantified against a dataset of Argo temperature and salinity profiles. The analysis was limited to a region around Australia, where the major difference between model versions was a change in vertical mixing scheme.

In both model versions, a deep bias existed in the region 20°S - 40°S and a shallow bias around 45°S - 65°S. The magnitude of the bias was decreased in version 3.0 and was nearly erased in the region 45°S - 50°S. A significant decrease in the



variability of MLD estimates between model forecast runs was attributed to a shorter hindcast cycle in version 3.0. Version 3.0 also outperformed version 2.2.1 in all regions in terms of skill versus climatology. Skill versus persistence was also investigated but results were inconclusive; it is likely that additional sources of error are introduced into the persistence forecast included in the dataset by combining independent model cycles.

In nearly all areas, the magnitude of the mean absolute error (MAE) was reduced in version 3.0. The only exceptions were seen in the region 20 – 30°S, 90 – 100°E; where a weakly stratified mixed layer is common, and south of 55°S, where convective overturning is a significant mixing mechanism. These results suggest that while in most instances version 3.0 outperformed version 2.2.1, in situations where a positive (negative) buoyancy flux is a significant factor in making the mixed layer more (less) stable, the GOTM mixing scheme may generate excessive (insufficient) vertical mixing.

Having discounted other factors, it is most likely that significant improvements in the estimation of the MLD are due to the change from the Chen et al. (1994) mixing scheme in version 2.2.1 to the GOTM in version 3.0. While limitations in the calculation of buoyancy production have been noted in version 3.0, the rectification of this issue is expected to deliver further improvements in mixed layer representation for future iterations of the OceanMAPS model.

**Data availability**

The Class 4 dataset and the analysis code used in this study is available at

http://handle.unsw.edu.au/1959.4/resource/collection/resdatac_530/1

**Competing interests**

The authors declare that they have no conflict of interest.

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



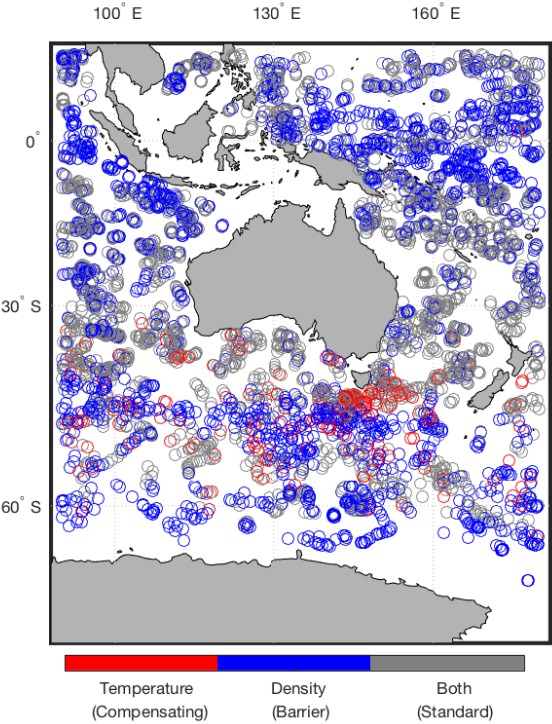

**Figure 1. Criterion used the identify the MLD for each Argo observation. Use of the density criteria implies the existence of a barrier layer, while use of the temperature criteria implies the existence of a compensated layer.**



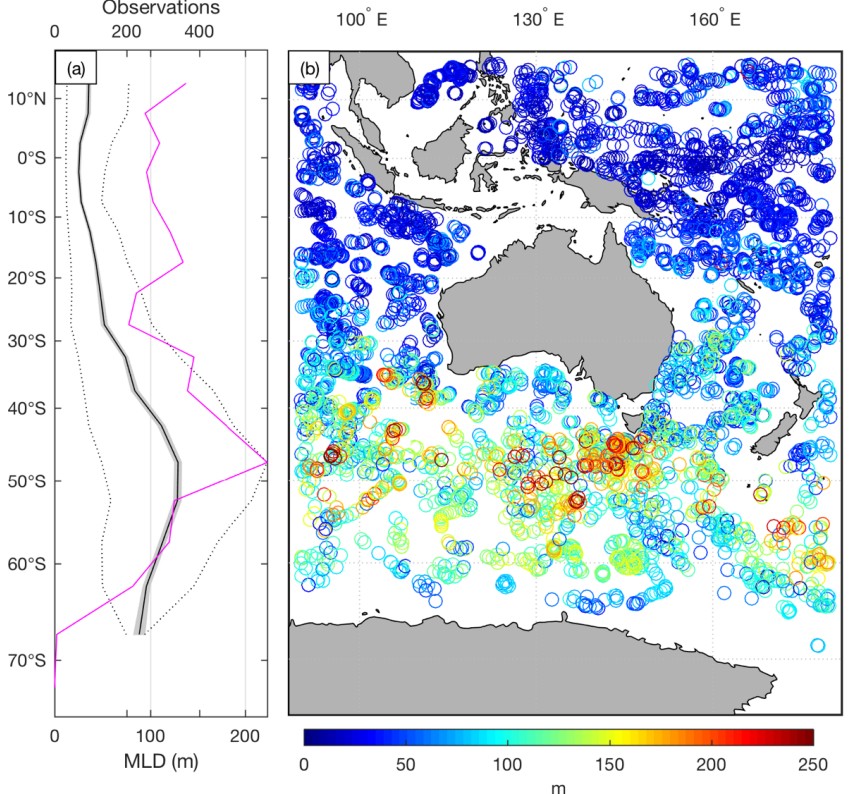

**Figure 2. (a) The zonal mean of the observed MLD (m) derived from Class 4 Argo profiles over the study period; 90% confidence intervals are shaded and the 5th and 95th percentiles are shown by the dotted lines. The number of profiles in each 5° bin is also indicated by the magenta line. (b) The individual MLD observations.**

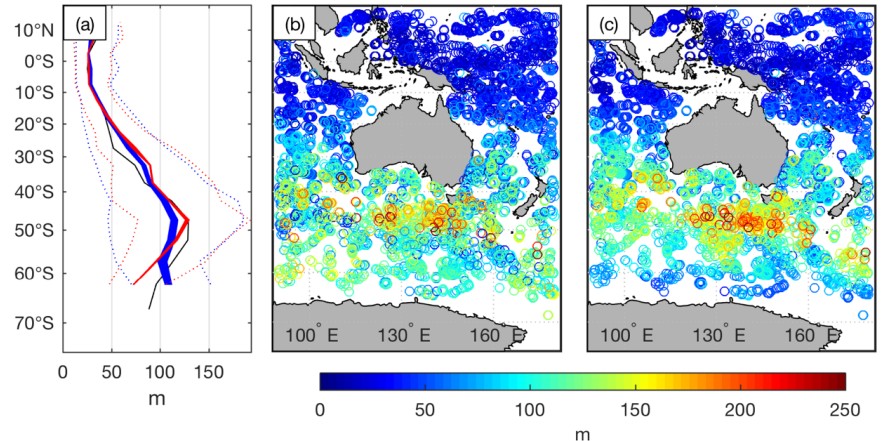

**Figure 3. (a) The zonal mean of the OceanMAPS version 2.2.1 (blue) and 3.0 (red) MLD corresponding to Class 4 profiles over the study period. The range of values between forecast time steps (+0 h, +24 h, +48 h etc) is indicated by the line thickness. The 5th and 95th percentiles for each model are shown by the dotted lines. The observed mean (from Fig. 2) is shown in black. The forecasts at model time +24 are shown at (b) for version 2.2.1 and (c) for version 3.0.**




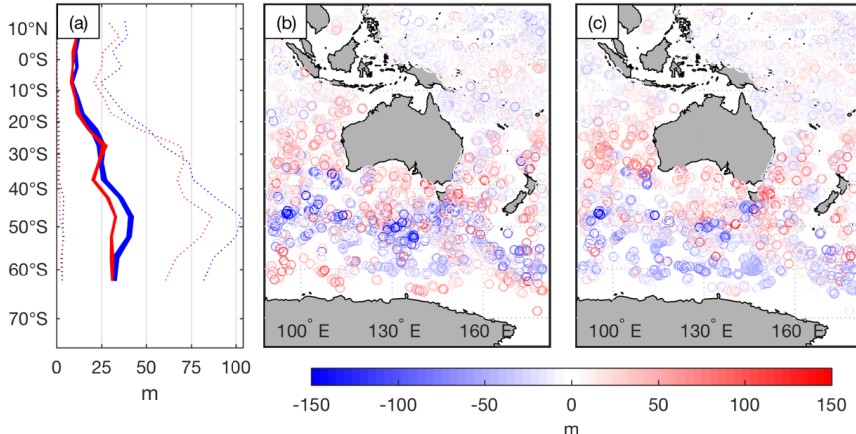

**Figure 4. (a) The mean absolute error (MAE) of the OceanMAPS 2.2.1 (blue) and 3.0 (red) MLD corresponding to Class 4 profiles over the study period. The range of values between forecast time steps (+0 h, +24 h, +48 h etc) is indicated by the line thickness. The 5th and 95th percentiles for each model are shown by the dotted lines. The observed mean (from Fig. 2) is shown in black. The MAE at model time +24 are shown at (b) for version 2.2.1 and (c) for version 3.0.**

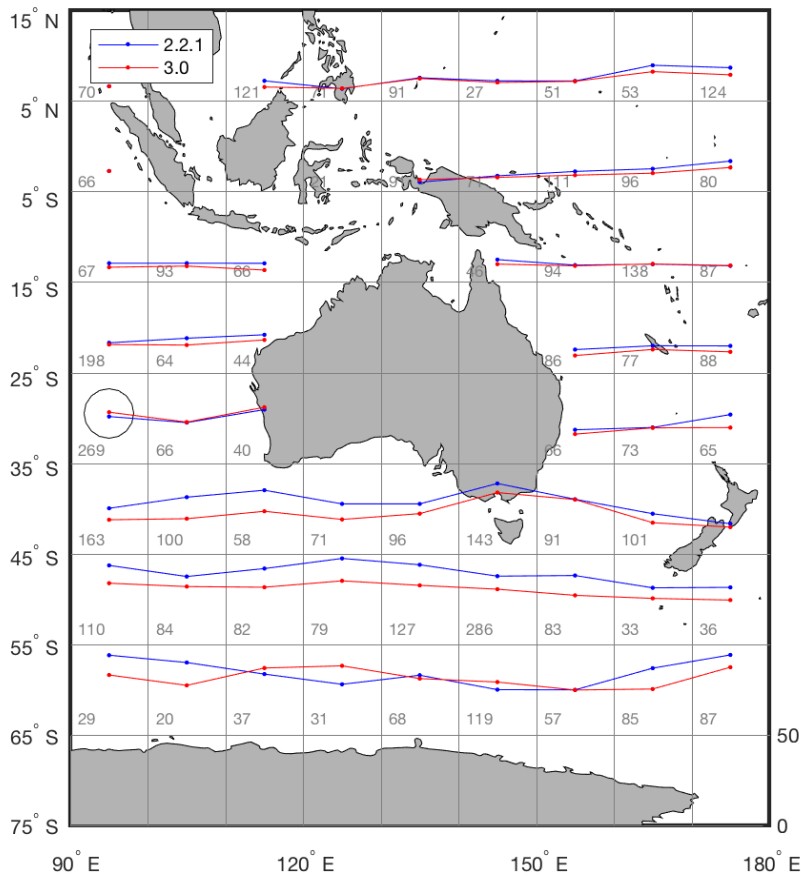

**Figure 5. Mean absolute error (MAE) for OceanMAPS version 2.2.1 (blue) and 3.0 (red), averaged over 10° × 10° latitude and longitude bins. Each plot is scaled over 0 – 50 m. The number of profiles within each bin is indicated by the grey text. The area where version 2.2.1 performs better than version 3.0 discussed in section 0 is circled.**





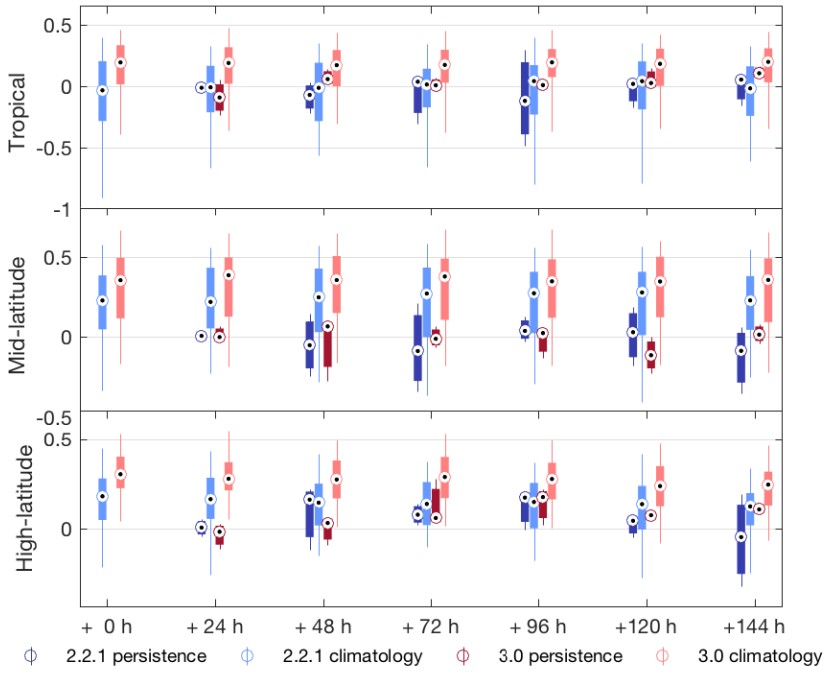

**Figure 6. Model climatology skill and persistence skill for OceanMAPS 2.2.1 (blue) and OceanMAPS 3.0 (red). The 5<sup>th</sup> and 95<sup>th</sup> percentiles are shown by the whiskers, while the middle quartile is shown by the boxes. Data has been binned into tropical (16°N to 20°S), mid-latitude (20°S to 45°S) and high latitude (45°S to 75°S) regions.**



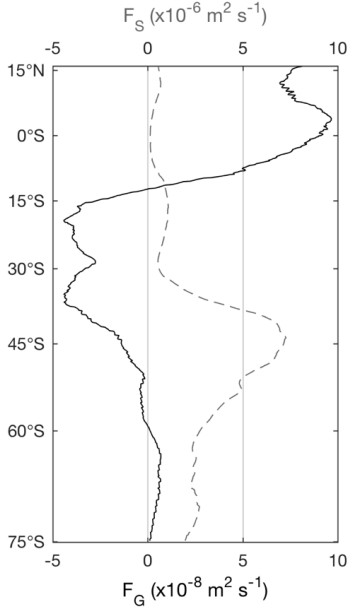

**Figure 7. The zonal mean surface buoyancy ($F_G$, solid line) and shear ($F_S$, dashed line) flux from OceanMAPS version 3.0. A positive (upwards) buoyancy flux tends to make the mixed layer unstable and promote mixing.**

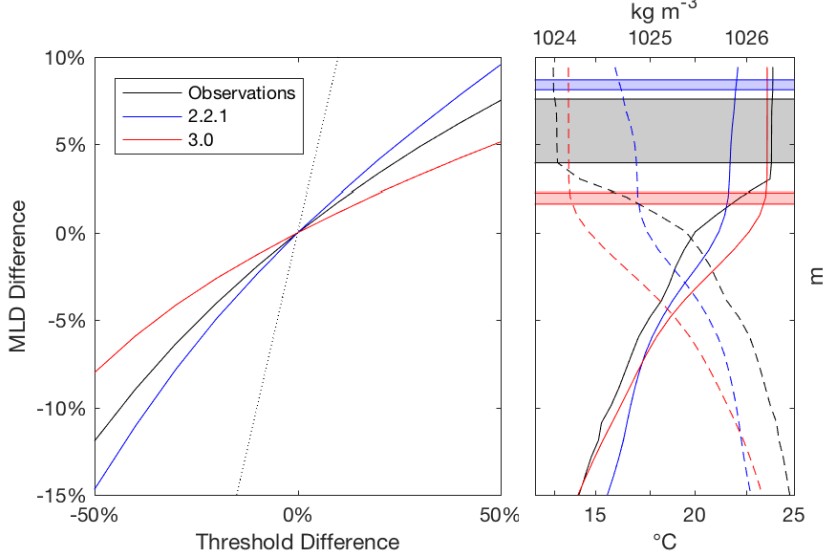

5   **Figure 8. (left) Mean difference in the calculated MLD as a result of varying the magnitude of the temperature and density thresholds. The dotted line indicates a slope of unity. (right) A typical Argo temperature (solid) and density (dashed) profile (location 28°S 96°E), with the corresponding model estimates. The relative sensitivity of each dataset is indicated by the range of MLD estimates (shaded).**



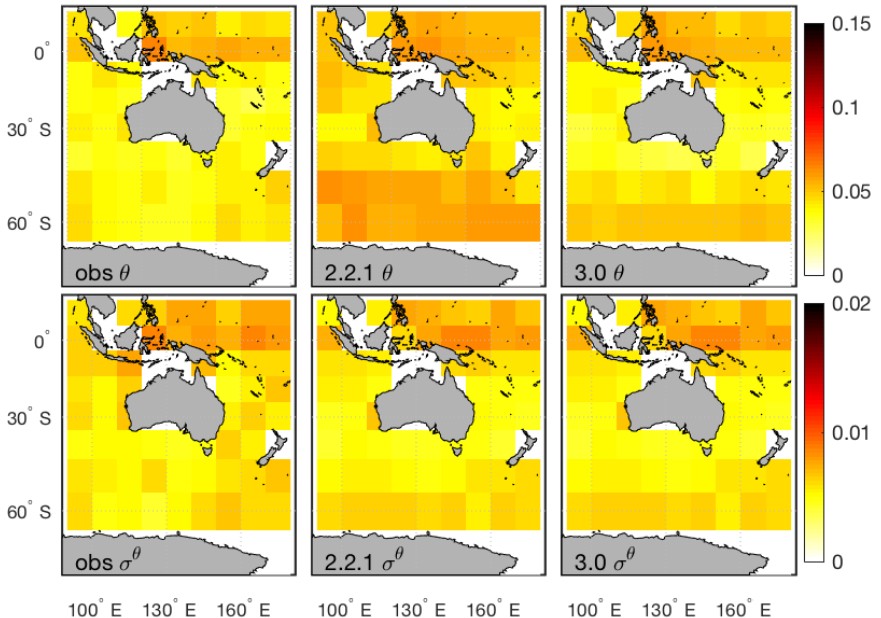

**Figure 9. The mean standard deviation of the potential temperature (top) and potential density (bottom) above the MLD, for observations (left), OceanMAPS version 2.2.1 (centre) and version 3.0 (right).**

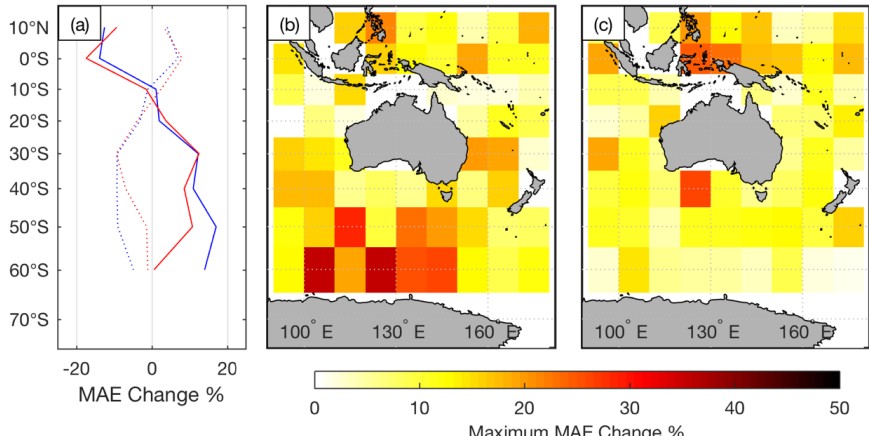

5 **Figure 10. (a) The percentage change of the MAE in MLD when the temperature and salinity thresholds are varied by +50% (dotted) and -50% (solid), for OceanMAPS version 2.2.1 (blue) and 3.0 (red). The maximum absolute change in MAE recorded for a ± 50 % change in thresholds is shown in 10° × 10° latitude and longitude bins for version 2.2.1 (b) and 3.0 (c) are also shown.**



**Table 1. Details of the OceanMAPS versions compared in this study**

|  | *Version 2.2.1* | *Version 3.0* |
|---|---|---|
| *Operational* | 10 Nov 13 | 14 Apr 16 |
| *Domain* | 0 – 360°E, 75°N – 75°S ||
| *Horizontal resolution* | 0.1° (90°E - 180°E, 16°N - 75°S)<br>0.1° – 2.0° elsewhere | 0.1° |
| *Vertical resolution* | 5 m (0 – 20 m)<br>5 – 10 m (20 – 90 m)<br>> 10 m (below 90 m) ||
| *Data assimilation* | BODAS (Oke et al., 2008;Andreu-Burillo et al., 2010) | EnKF-C (Sakov, 2014) |
| *Forecast scheduling* | Forecast: 4 independent models run on consecutive days<br>Near real-time analysis: -5 to 0 days<br>Behind real-time analysis: -9 to -5 days | Forecast: 3 independent models run on consecutive days<br>Near real-time analysis: -3 to 0 days<br>Behind real-time analysis: -6 to -3 days |
| *Forecast period* | 0 – 144 h ||
| *Atmospheric fluxes* | Surface wind stress, shortwave radiation, longwave radiation, sensible heat flux, evaporation and precipitation from the ACCESS-G model (Puri et al., 2013) ||
| *Vertical mixing* | Chen et al. (1994) modified by Power et al. (1995) | GOTM (Burchard et al., 1999) configured as $k$-$\varepsilon$ |
| *Topography* | Smith and Sandwell version 11.1 (Smith and Sandwell, 1997) | 9' around the Australian region (Whiteway, 2009) and the 30' GEBCO 08 (BODC, 2008) elsewhere |
| *River runoff* | Based on global climatology (Dai and Trenberth, 2002) ||