# Peer review of "Verification of the mixed layer depth in the OceanMAPS operational forecast model for Austral autumn"

_Geoscientific Model Development, 2018_

## Referee Comment (RC1) · Anonymous Referee #1 · 1 Jun 2018

General comments:

This is a good paper. It is a regional comparison between two different models utilizing a local contribution to a major community project (GODAE OceanView Class-4 Inter-comparison) as part of the validation of a model upgrade. I found the explanations and figures to be clear and concise, with sufficient background on the methods of MLD estimation as to carry the discussion without necessitating an immediate literature search. It also used subregional groupings (latitudinal zones) that helped highlight the differences between the two MLD algorithms in the two models and how they compared to in-situ observations.

Scientific comments

I am a little unsure of the purpose of section 5.2, the sensitivity study based on modifying the accepted MLD threshold definition. In particular, this sentence stands out: "Clearly an optimal MLD definition may include spatially varying thresholds; however, unless carefully implemented this may introduce other errors." To my understanding, the MLD definition used in the paper (de Boyer Montégut et al., 2004) is an optimal definition, calibrated on a global scale on a 2-degree grid. The validation study is dependent on using consistent community-accepted metrics, as is the GODAE Ocean-View data set it uses. Suggesting that there may be a regional issue with the metric definition may be outside of the scope of the paper, or possibly the subject of another, more in-depth study. At the very least the statement needs support from the literature.

A general comment on the MLD statistics: the de Boyer Montégut 2004 paper suggested that "Comparisons with ocean models should be made with MLDs that are computed at each timestep and then averaged, yielding more consistent comparisons." Considering that the validation study uses the GOV data set which is comprised of daily averaged profiles from which the MLD is then computed, this would appear to present something of a problem. Possibly some discussion about the particulars of the GOV data set construction, perhaps a comparison of daily averages of instantaneous MLD versus MLD from daily averaged profiles would strengthen the discussion. To my understanding, when calculating statistics from non-linear processes, it's best to compute the statistic from the data and then average the results, rather than average the data and then compute the statistic.

Technical comments

Section 4.3 line 16-17: "Both models are more skilful than climatology within that data limits (+144 h), and in each region, there is a distinct improvement in version 3.0 compared to 2.2.1." Not sure if "skilful" ought to have two ells (skillful), or if its a regional usage. Also, "that" appears to be a typo. In any case the sentence is not clear. Suggest saying something like this: "Both models are more skillful than climatology within their respective data limits out to 144 hours, and within each region there is a distinct

[Figure]

improvement in version 3.0 as compared to version 2.2.1."

Figure 4(a): The observed mean line (shown in black) is not evident in the image, due to the close overlap of the blue MAE line. The figure is relatively small, and blue/black differences are slight and difficult to perceive. Suggest changing the MAE color to something brighter, possibly using a dashed line for the overlapping MAE line to allow the mean line to show through.

Figure 5: A label on the secondary y-axis for the MAE would greatly help the graphic. It's mentioned in the caption, but a label would clearly identify the axis values.

Figure 6: The contrast between the light and dark red/blue stripes is not very strong (though that may be a product of my printer calibration). Perhaps more contrasting colors for the respective climatologies? This is more of a minor stylistic opinion on my part.

––––––––––––––––––––––––

---

## Referee Comment (RC2) · Anonymous Referee #2 · 16 Jun 2018

general comments

The presented study proposes an evaluation, during the autumn (April-July 2016), around Australia, of the forecast Mixed Layer Depth (MLD) simulated both by the new and old version of the operational forecast systems of Australian Bureau of Meteorology. To validate, the authors use data developed in the framework of the GODAE Ocean View program. They propose a very precise description of different oceanic stratification cases that can induce changes in the calculation of MLD. Then, both systems are evaluated on their ability to predict the MLD. To finish, a discussion is proposed around the impact of the model changes and around definition of the MLD and its sensitivity to the thresholds. I have found this paper very interesting with very good point as the choice of the data to validate (a shared data base). This give the possibility to others groups reimplementing the proposed diagnostics. The particular attention to describe the problematic of the MLD determination (the different cases depending on the temperature and salinity variations) is convincing and very well explained. The quality of the graphs is good. My major comments are i) on the very short period used to perform the study compare to some affirmative sentences (and title of the paper) ii) for most of the time, the attribution of the observed changes to the vertical mixing scheme without clear proofs. (A.1) About the first point, I understand this prototypes are huge and running long tests is quite impossible. But, I think in these cases, it is not correct speaking of verification without mentioning the study period. This study occurs in austral autumn and stratification/destratification events will be completely different compare to other seasons. The authors should moderate some affirmations, which could change during other periods, and precise more often the study period as in conclusion, title, ... (A.2) About the second point, I aslo agree that the vertical mixing scheme should play a major role but regarding the list of changes between both version of systems (assimilation method, cycles of assimilation, mixing scheme, bathymetry, horizontal grid, version of the code ?), I think it's quite impossible giving strict affirmations on its major or minor impact. All the listed changes mention in table 1 are major. The authors should use the words of "we suppose", "we suspect",... On this attribution of the changes, the authors give some justification but not always convincing. See the specific comments in the following part for more details (B.1).

specific comments

(B.1) Some discussion to attribute the major part of observed change to the vertical mixing are not enough convincing. The authors should moderate some affirmations. For example, page 6 line 5-10, the paragraph that explains the new grid has no impact, is not so clear for me. The authors base their argumentation on MAE and not on normalized values. On the figure 2, we observe that the observed MLD is zonal. For this reason, I think this is normal to have the same zonal pattern on MAE even if the grid changes. But a MAE of 20m is not the same for an MLD of 200m than for an
MLD of 50m. A map with, percentage of error should be more convincing. Or page 7 line 10-14, I think the affirmation is too strong. The Figure 6 present the skill of both systems. In each bands, the new system performs better than the old one at T+0h. I think, it demonstrates the importance of the change of the cycles and data assimilation scheme. Therefore, the profiles presented in figure 8 or figure 9 could be a consequence of the "assimilation work" and not of the vertical mixing scheme ?

(B.2) For model description, the authors report to (Oke et al., 2013). Despite I think a short description of the vertical grid could be useful because it is a model characteristic of prime importance for the representation of the MLD.

(B.3) In (Oke et al., 2013), the presented simulation is performed with (Chen et al. 1994) algorithm. I wonder why it was decided to change the vertical scheme in the operational system and why you have chosen k-epsilon formulation. Have you performed some studies with idealized, 1D or with lower resolution model as for example in Reffray et al. 2015 ?

(B4) The vertical mixing scheme is briefly described. Could you add some information as: i) Have you some modifications/parameterizations of GOTM tacking into account the effect of waves (change of roughness depending on wind field, mixing induced by breaking waves)? A such parameterization could have a strong impact on vertical mixing, especially in ACC. ii) On the possible impact if stability functions are included (see for example Burchard and Bolding, 2001)

(B5) In this paper, a few words about the possible impact of a Sea Ice model would have been interesting. Somme comments about this point would be interesting, especially the impact during the austral winter in the south of the domain ?

(B.6) Have the authors evaluated the changes induced by the modification of the bathymetry data base and/or the grid? The circulation in the north of the study domain is strongly driven by the straits and their representations in the model. If there are changes in the circulation, it could explain differences particularly in the North-West of

the domain (outflow coming from Indonesia).

(B.7) The affirmation page 5 line 10 (RMSE between 0.2-0.7) needs a reference.

technical corrections

(C.1) in legend Figure 5 there is a report to a "section 0" ?
* * *

---

## Author Comment (AC1) · 18 Jul 2018

**1 Scientific comments**

On a spatially dependent MLD definition. The reviewer questions our suggestion that the suitability of the MLD criteria thresholds may vary spatially. We adopted de Boyer Montegut et al., (2004) as a well performing and widely cited scheme. However, there are a number of clear points that would not support this as an optimal definition. As we discuss in section 3, the calculation of the MLD is complicated by the wide variety of temperature and density profiles observed in the ocean, making any MLD criteria susceptible to errors in certain circumstances. The temperature criterion is not set to be optimal in every global location or season but a single criterion that is "fairly successful"

[Figure]

at estimating the MLD. The principal justification for the threshold is based on the representation of the spring re-stratification of observations in Arabian Sea and Subpolar North Pacific. This is well demonstrated in their Figure 4b and 4d. Furthermore, the 0.2 degC threshold was compared with 0.5 and 0.8 degC as previously cited values. However, by making use of the 10 m reference temperature a smaller threshold is made possible as clearly shown. However refinement of the criterion in the range 0.1 to 0.3 degC is not made. Figure 4c in their paper shows the criterion underestimates the MLD in Sub-tropical North Pacific and a value between 0.2 and 0.5 degC is likely best for this region. How this impacts the other two regions would need to be investigated. We argue that the method as described has an uncertainty of at least +/- 50%. A similar range of uncertainty for the density thresholds is to be found in de Boyer Montegut et al., (2004). In addition, our study examines a period of Austral autumn and the subtleties of spring re-stratification are absent. The MLD estimates from the observations should be robust to this change in threshold, though we find a nonlinear response. More important; however, is that we are using the observed sensitivity as a reference to interpret the response of the model. In any event, we have modified this paragraph to more clearly outline our argument.

On MLD statistics. The reviewer has suggested that it would be more advantageous to compare instantaneous OceanMAPS profiles to the Argo observations. While we agree with this suggestion, the standard OceanMAPS forecast output only provides 24 hr mean fields for sub-surface variables. As we do not have the resources to reproduce the OceanMAPS operational forecasts, higher resolution time steps are not available. In addition, the 0.1 deg resolution of the model fields introduces spatial averaging errors that are not present in the Argo observations. However, given the randomness of the float location within a cell and the time relative to the centre of the time-average we do not expect this to lead to systematic errors. To remove the influence of diurnal variation that may be present in the near-surface Argo observations, we have defined our MLD criteria in terms of a reference depth of 10 m; below this depth negligible diurnal variation can be expected and the model mean profiles can be considered representative of the instantaneous fields. While this method does limit our MLD estimates to depths below 10 m, the vertical resolution of the model (only two levels above 10 m) would preclude any accurate estimate of such a shallow MLD. A more detailed explanation of the available OceanMAPS data has now been included in Section 2.2 of the dataset.

2 Technical comments

Section 4.3 line 16-17. 'Skilful' is the UK spelling, while 'skillful' is common in the USA and Canada (see https://en.oxforddictionaries.com/definition/skilful). We have retained the UK standard. The reviewer also identified a typographic error ("that"); we have adopted their recommended change.

Figure 4(a). On the absence of a mean (black) line. The reviewer noted that the observed mean line (in black) was not evident in this figure, despite being identified in the caption. This was a typographical error in the caption, and the mean line (from figure 2) was not intended to be included.

Figure 5. A label has been added to the y-axis legend

Figure 6. The contrast between the light and dark bars has been increased
* * *

---

## Author Comment (AC2) · 18 Jul 2018

**1 General comments**

A1 .On the temporal extent of the dataset. While the relatively short duration of the dataset had been mentioned in the manuscript, we have expanded upon the implications of this for our results and modified the title to include the season.

A2. On the impact of model changes on MLD estimation. The reviewer questioned whether the impact on MLD performance of other changes between model versions can be assumed to have a minor impact. We agree that it is not possible to completely isolate these other impacts, but our analysis was designed to minimise the impact of changes other than the vertical mixing scheme. We also note that the configuration of

the version 3.0 system was developed over a series of disparate hindcast experiments, over different time periods and geographical regions. Because of this it is not possible to quantify the impact of each change systematically, but we argue in section 2.1 that these changes will be of second-order effect in comparison to the vertical mixing scheme. For example, while the data assimilation software was upgraded, both versions utilise an identical ensemble optimal interpolation method (The software allows for EnKF, but this is to be implemented in a later version). Changes to the bathymetry are mostly limited to the continental shelf (discussed below), while our Argo dataset is concentrated over the deep ocean. The main change, apart from the vertical mixing scheme, is the reduced spin-up period in version 3.0. We minimise the impact that this will have on our results by performing the analysis over the forecast period, which is three to five days beyond the spin-up period. In any case, we acknowledge that our analysis does not constitute a controlled experiment and this poses limits to the attribution and interpretation of results. Where relevant, we have noted these limitations and modified our assertions as appropriate in the revised manuscript.

2 Scientific comments

B1. On the discussion of vertical mixing. Our argument regarding the impact of a telescopic grid has been restructured to make it clearer. We have also revised figures 4 and 5 to show normalised MAE as suggested by the reviewer.

B2. On the model description. Additional details regarding the model vertical coordinate have been included.

B3. On the reason for a change in vertical mixing scheme. When version 2.0 of Ocean-MAPS was implemented, the Chen et al. (1994) mixed layer scheme was acknowledged by its developers as out of date. However, testing with a KPP scheme did not show a significant reduction in errors and Chen et al (1994) was retained. The MOM4 model provided the GOTM package as an option which prompted the application of a K-epsilon scheme. This was first undertaken in regional studies with positive results

before being applied to the operational global model.

B4. On the implementation of the GOTM scheme. The implementation of GOTM in OceanMAPS version 3.0 includes the breaking wave TKE model of Craig and Banner (1994), as modified by Umlauf et al. (2003). This is an improvement on the Chen et al (1994) scheme, which did not explicitly account for breaking waves. However, while breaking waves have certainly been shown to increase mixing in the ocean boundary layer, the depth over which this occurs is of the order of the significant wave height (see e.g. Terray et al., 1999, D'Asaro, 2014, Umlauf et al., 2003). In the region of the ACC, where the MLD is O(100m) during the study period, it is unlikely that breaking waves would play a significant role in mixed layer deepening. We assess that the inclusion of a breaking wave parameterisation would have a small, but largely insignificant, impact on the version 3.0 MLD results in this study period. The stability functions of Schumann and Gerz (1995) was used in the GOTM implementation in OceanMAPS version 3.0. While use of a more complex function (e.g. as discussed in Burchard and Bolding, 2001) may improve MLD results, the primary aim of our study is to compare the performance of the two model versions and a discussion of the GOTM settings to this detail falls outside of this scope. A follow-on study conducted as a controlled experiment would be ideal to quantify the impact of these settings.

B5. On the impact of sea ice. The lack of a sea-ice parameterisation within Ocean-MAPS certainly affects its performance at high latitudes. However, as both versions of OceanMAPS compared in this study do not include sea-ice, the impact on the relative performance of each model is negligible.

B6. On the impact of the upgraded bathymetry. A change in bathymetry dataset for version 3.0 has mainly resulted in better resolution over the continental shelf with negligible change over the deep ocean (figure R1, below). The coverage of the CLASS4 dataset is negligible over the continental shelf (figure 1). As tidal forcing is not included in OceanMAPS, the impact of bathymetry on internal tide mixing also does not affect a comparison between model versions.
B7. On skill scores. References to typical skill scores have been included in the manuscript.

3 Technical comments

C1. The reviewer identified an error in the document cross-referencing. This has been corrected

**Fig. 1.** Relative bathymetry change from OceanMAPS version 2.2.1 to version 3.0